# Prognostic Value of Pulmonary Artery Pulsatility Index in Right Ventricle Failure-Related Mortality in Inoperable Chronic Thromboembolic Pulmonary Hypertension

**DOI:** 10.3390/jcm11102735

**Published:** 2022-05-12

**Authors:** Sylwia Sławek-Szmyt, Aleksander Araszkiewicz, Stanisław Jankiewicz, Marek Grygier, Tatiana Mularek-Kubzdela, Maciej Lesiak

**Affiliations:** 1st Department of Cardiology, Poznan University of Medical Sciences, 61-848 Poznan, Poland; aaraszkiewicz@interia.pl (A.A.); stanislaw.jankiewicz@skpp.edu.pl (S.J.); mgrygier@wp.pl (M.G.); tatiana.mularek-kubzdela@skpp.edu.pl (T.M.-K.); maciej.lesiak@ump.edu.pl (M.L.)

**Keywords:** chronic thromboembolic pulmonary hypertension, survival, risk stratification, right ventricular failure, mortality

## Abstract

Chronic thromboembolic pulmonary hypertension (CTEPH) is an ominous disease leading to progressive right ventricular failure (RVF) and death. There is no reliable risk stratification strategy for patients with CTEPH. The pulmonary artery pulsatility index (PAPI) is a novel hemodynamic index that predicts the occurrence RVF. We aimed to investigate prognostic value of PAPI in inoperable CTEPH. Consecutive patients with inoperable CTEPH were enrolled. PAPI was calculated from baseline right heart catheterization data. A prognostic cut-off value was determined, and characteristics of low- and high-PAPI groups were compared. The association between risk assessment and survival was also evaluated. We included 50 patients (mean age 64 ± 12.2 years, 60% female). The number of deaths was 12 (24%), and the mean follow-up time was 52 ± 19.3 months. The established prognostic cut-off value for PAPI was 3.9. The low-PAPI group had significantly higher mean values of mean atrial pressure (14.9 vs. 7.8, *p* = 0.0001), end-diastolic right ventricular pressure (16.5 vs. 11.2, *p* = 0.004), and diastolic pulmonary artery pressure (35.8 vs. 27.7, *p* = 0.0012). The low-PAPI group had lower survival as compared to high-PAPI (log-rank *p* < 0.0001). PAPI was independently associated with survival and may be applicable for risk stratification in inoperable CTEPH.

## 1. Introduction

Chronic thromboembolic pulmonary hypertension (CTEPH) is a relatively rare disease characterized by intravascular macroscopic thromboembolic lesions and microscopic pulmonary vascular remodeling. The consequence of these pathological changes is increased pulmonary vascular resistance (PVR), leading to right ventricular failure (RVF) and death if left untreated [1,2,3].

A pulmonary endarterectomy (PEA) is the treatment of choice for operable patients. However, some patients are ineligible for surgery due to distal localization of thromboembolic lesions, comorbidities, or functional status [4,5]. Available treatment options for these CTEPH patients include targeted medical therapy and interventional technique—percutaneous balloon pulmonary angioplasty (BPA), which has become a promising treatment modality [6,7,8,9,10,11,12].

Currently, there is no established risk assessment tool guiding treatment decisions in patients with inoperable CTEPH. Although the assessment tool proposed by the European Society of Cardiology (ESC) and European Respiratory Society (ERS) for assessing mortality in pulmonary arterial hypertension (PAH) has also been also evaluated in CTEPH patients, there are no reliable tools dedicated for patients with inoperable CTEPH [13,14,15,16].

The pulmonary artery pulsatility index (PAPI) is a novel hemodynamic parameter designed to assess the ability of the right ventricle (RV) to generate a pressure gradient and is defined as difference between systolic pulmonary arterial pressure (sPAP) and diastolic pulmonary arterial pressure (dPAP) in the nominator divided by mean right atrial pressure (mRAP) in the denominator ((sPAP-dPAP)/mRAP) [17]. Some previous studies revealed that PAPI predict RVF development in patients with acute inferior myocardial infarction and after left ventricular assist device implantation in end-stage left heart failure [17,18,19]. We hypothesized that PAPI might be also predictive marker in inoperable CTEPH patients. Therefore, the aim of the present study was to evaluate whether PAPI might be useful for mortality prediction in inoperable CTEPH.

## 2. Materials and Methods

### 2.1. Study Population

We consecutively enrolled all patients diagnosed with inoperable CTEPH (World Health Organization [WHO] group 4 of pulmonary hypertension) in our center between January 2015 and December 2019. CTEPH was confirmed when mean pulmonary arterial pressure (mPAP) was greater than 25 mmHg at rest and pulmonary artery wedge pressure (PAWP) was lower than 15 mmHg measured directly during right heart catheterization, and perfusion defects were revealed by conventional pulmonary angiography and computed tomography pulmonary angiography (CTPA) after at least 3 months of optimal anticoagulation therapy [13]. Other secondary causes of pulmonary hypertension were excluded by appropriate lab tests and imaging modalities according to the current guidelines of the European Society of Cardiology (ESC) [13].

The baseline assessment included demographics, personal medical history, World Health Organization functional class (WHO-FC), six-minute walking distance (6MWD), arterialized capillary blood gasses, N-terminal pro-B-type natriuretic peptide (NT-proBNP) concentration, lung function tests, echocardiographic, angiographic, and hemodynamic results.

All patients with CTEPH were discussed by a multidisciplinary team consisting of a clinical cardiologist experienced in treatment of pulmonary hypertension, interventional cardiologist experienced in BPA, and PEA cardiac surgeon. CTEPH patients were regarded as inoperable when either distal, surgically inaccessible vascular occlusions were present and further qualified for targeted medical treatment and sequential BPA procedures. Each patient with inoperable CTEPH received medical therapy with soluble guanylate cyclase stimulator, riociguat—the only drug approved for treating CTEPH. Riociguat was initiated within 3 months of diagnosis. Moreover, patients were offered BPA procedures with detailed information about the potential risks and benefits of this intervention, and each patient underwent the series of BPA in our center.

### 2.2. Right Heart Catheterization and Pulmonary Angiography

Right heart catheterization (RHC) was performed via the right internal jugular vein or right common femoral vein access using a flow-directed, balloon-tipped Swan–Ganz catheter (7F; Edwards Lifesciences, Irvine, CA, USA) in a supine position according to current guidelines [20]. The following pulmonary circulation parameters were directly measured at end-expiration or calculated: right atrial pressure (RAP, systolic, diastolic, and mean (mRAP)), right ventricular pressure (RVP, systolic (sRVP), diastolic (dRVP) and end-diastolic (edRVP)), pulmonary arterial pressure (PAP, systolic (sPAP), diastolic (dPAP) and mPAP), pulmonary arterial wedge pressure (PAWP), cardiac output (CO), cardiac index (CI), stroke volume (SV), pulmonary vascular resistance (PVR), and mixed venous saturation (SvO_2_). Cardiac output was evaluated using the thermodilution method. Pulmonary vascular resistance was determined as the difference between mPAP and PAWP divided by CO [17]. The PAPI was calculated as:PAPI = sPAP − dPAP/mRAP

The detailed description of pulmonary angiography and CTPA procedures at our center have been previously published [21].

CTEPH was categorized as proximal (lesions predominantly located in the main, lobar, and proximal segmental arteries) or distal (lesions distributed in distal segmental, subsegmental, or more distal vessels with remodeling of microcirculation) in accordance with San Diego intraoperative CTEPH classification [22]. 

### 2.3. Echocardiography

Transthoracic echocardiography was performed to qualitatively assess right ventricle strain parameters: RV free wall thickness, RV end-diastolic diameter, tricuspid annular plane systolic excursion (TAPSE), S’ wave, and degree of tricuspid regurgitation.

### 2.4. Risk Stratification

In accordance with the 2015 ESC/ERS guidelines risk assessment tool, patients were categorized as low, intermediate, or high risk based on variables: WHO FC, 6MWD, NT-proBNP, RA area, presence of pericardial effusion, mRAP, CI, and SvO_2_ [13]. Every risk variable in each patient was graded from 1 to 3, where 1 was low risk and 3 was high risk, and the sum of the points was divided by the number of variables, and the result defined the risk of the individual patient as described previously [14].

### 2.5. Statistical Analysis

Patients’ characteristics are expressed as frequency (percentage) for categorical variables and mean ± standard deviation or median and interquartile range for continuous variables, as appropriate. The normality of distribution was assessed using the Shapiro−Wilk test and Lilliefors test. Categorical variables were compared using the two-tailed Fisher’s exact test or χ^2^-test. Differences between continuous variables were tested with the Student’s *t*-test, the Mann−Whitney test, or Wilcoxon test as appropriate. PAPI calculated from hemodynamic data was categorized into “low” and “high” groups with the chi-square automated interactions detector algorithm as described previously [19,23]. This is a multi-way splitting decision tree used for identifying the predictors or explaining an outcome based on the adjusted Bonferroni testing or determining the optimal cut-off values for the predictors. The discretization would enable one to examine the underlying patterns of association between PAPI and the outcomes. The generalized structural equation model was used for confirmatory analyses, with Binomial and Weibull chosen as the underlying distributions for assessing the occurrence of death (1: yes; 0: no) and time to death (months), respectively. Beyond the PAPI, the predictors also involved demographics, clinical data, and other hemodynamic parameters. The results were presented based on a backward elimination procedure in model-selection, which was in turned based on the unadjusted analyses. The accuracy of the derived PAPI cut-off was subsequently evaluated with the receiver-operating characteristics (ROC) curve. The ROC analysis was also performed to assess discriminative capacity of potential survival predictors. Survival was analyzed both at baseline and follow-up using the Kaplan–Meier method with log-rank test. Predictors for survival were determined with univariable and multivariable Cox proportional regression analyses. Hazard ratios with 95% confidence intervals are presented. The alpha significance level of 0.05 was set up. Statistical analysis was performed using Statistica version 13.7 (StatSoft, Inc., Tulsa, OK, USA).

## 3. Results

### 3.1. Study Population

We included 50 out of 70 patients from our registry after excluding 20 patients with lack of informed consent, CTEPH persistent after PEA, non-CTEPH related death or incomplete follow-up (see Appendix A). The number of deaths was 12 (24%), and the mean follow-up time was 52 ± 19.3 months (range: 7–84 months). The baseline demographic and clinical characteristics of the study cohort are provided in Table 1. The mean age was 64 ± 12.2 years, and 60% of the patients were female. Most patients were in WHO-FC III (56%) at initial assessment.

### 3.2. Risk Categorization Based on Pulmonary Artery Pulsatility Index

The distribution of PAPI values is shown on Figure 1. The median PAPI value in the whole study population was 6.0 (IQR: 4.62–8.23).

The established prognostic cut-off value for PAPI was 3.9. Patients were divided into two groups according to the PAPI scores: low PAPI, which is lower than 3.9 (high-risk group), and high PAPI, which is equal or higher than 3.9 (low-risk group), respectively. The low-PAPI group had higher values of body mass index and more frequently suffered from chronic renal insufficiency as compared to high PAPI. There were also significant differences in 6MWD and WHO FC at initial assessment between PAPI groups. However, PAPI groups were comparable in terms of age, gender, as well as other comorbidities. PAPI groups were also similar regarding estimated ESC 2015 risk category and NT-proBNP concentrations (see Table 1).

With reference to hemodynamic parameters, the low-PAPI group had significantly higher mean values of mRAP (14.9 vs. 7.8, *p* = 0.0001), edRVP (16.5 vs. 11.2, *p* = 0.004), and dPAP (35.8 vs. 27.7, *p* = 0.0012). Notably, both PAPI groups were comparable in terms of sPAP, mPAP, PVR, CO, CI, and SvO_2_. Regarding to echocardiographic assessment, the low-PAPI group had significantly higher mean RAA and pulmonary trunk diameter. Other echocardiographic parameters were similar in both groups. Details are presented in Table 2.

### 3.3. Survival Analysis

As shown on Figure 2, survival was significantly lower low-PAPI group as compared with high-PAPI group (log-rank *p* < 0.0001). In the low-PAPI group, the survival rates were as follows: 80% (24 moths), 50% (36 months), and 30% (60 month), respectively. However, in the high-PAPI group, patients obtained survival rates of 97.5% (24 moths), 92.5% (36 months), and 82.5% (60 months), respectively. Differences in the survival rate between PAPI groups remained statistically significant during the follow-up period.

### 3.4. Mortality Risk Assessment

Receiver-operating characteristics curves for RVF-related mortality prediction were also analyzed. PAPI with the cut-off value of 3.9 was identified as useful risk mortality predictor (area under the curve (AUC) 0.83; CI: 0.7–0.97, *p* < 0.0001) along dPAP with the cut-off value of 31 mmHg (AUC 0.86; CI: 0.75–0.96, *p* < 0.0001) and with mRAP with the cut-off value of 10 mmHg (AUC 0.75; CI: 0.58–0.91, *p* = 0.0037). The ESC 2015 risk categorization was found to possess more limited discriminative ability than PAPI with AUC 0.71 (CI: 0.54–0.88, *p* = 0.017). Of note, sPAP, PVR, and age were not found to be significant predictors of mortality. Details are displayed on Figure 3. 

Table 3 illustrates the Cox proportional hazard regression model for prediction of RVF-related mortality in patients with inoperable CTEPH. The final optimized model included PAPI, 6MWD, and RAA.

### 3.5. Change in PAPI Values after Interventional Treatment of CTEPH

Change in PAPI values after implementation of targeted medical therapy and serial BPA procedures are shown on Figure 4.

## 4. Discussion

To the best of our knowledge, this is the first study to evaluate the prognostic utility of PAPI in a cohort of patients with inoperable CTEPH. The idea of using PAPI as a predictor for RV failure and mortality in inoperable CTEPH seems attractive due to simplicity of estimation and proven utility in other disease entities, such as acute inferior wall myocardial infarction, end-stage left heart failure, or pulmonary arterial hypertension (PAH) [17,18,19,24]. The numerator of the PAPI is defined as PA pulse pressure (PAPP), which reflects the combined of RV contractile function and pulmonary artery capacitance [24]. The denominator of the index is defined by mRAP, which represents RV preload [25]. The combination of these parameters into PAPI indicator provides insight not only into RV loading conditions but also mechanics. 

Nevertheless, the determined threshold for PAPI and its prognostic accuracy differ significantly between patients’ populations. In patients with RV myocardial infarction, PAPI equal or lower than 0.9 indicated an increased risk of RV mechanical support and mortality [17]. Following left ventricular assist device implantation, the optimal cut-off value for RVF development variers between studies from 1.85 through 2.0 up to 3.33 [18,25,26,27]. However, in PAH population (group I of PH according to ESC classification), a prognostic cut-off value was established between 3.7 and 5.3 [19,24]. In our inoperable CTEPH population, the derived PAPI cut-off value for mortality prediction is 3.9. The difference in PAPI thresholds and its predictive ability between various RVF etiologies seems to be related to the physiologic basics of PAPI. Pulmonary artery pulse pressure, the numerator of PAPI, is defined as RV stroke volume indexed against pulmonary arterial capacitance (PAC). PAC is negatively correlated with PVR. Hence, an increase of stroke volume and PVR raises pulmonary artery pulse pressure [19,28]. The pathophysiology of a disease needs to be taken into consideration in PAPI interpretation. CTEPH develops gradually due to chronic exposure to increased PVR and PAP (RV afterload), resulting in progressive RV failure. Right atrial pressure also increases gradually due to resultant compensatory grow in RA compliance [19,28]. Therefore, the PAPI threshold is higher in pulmonary hypertension. 

Our results showed the utility of PAPI for discrimination of mortality in patients with CTEPH. However, we identified several other parameters with reasonable prognostication ability, including mRAP, dPAP, and ESC 2015 risk assessment tool in ROC analysis. Previous studies also reported the significant correlation between mRAP as well as mPAP and PVR at baseline with survival in CTEPH patients [29,30]. Nonetheless, PVR and mPAP were not significant for prediction of mortality in our study. It was previously indicated that dPAP is less sensitive to flow metrics as compared to mPAP and was therefore proposed for the evaluation of diastolic pulmonary gradient (DPG = dPA − mean PAWP) as a precapillary component in patients with postcapillary PH [31,32]. Recently, Apitz et al. reported dPAP and dPAP-derived variables being useful parameters for hemodynamic assessment of pulmonary vascular disease, with similar reliability as mPAP and mPAP-derived parameters in children with idiopathic pulmonary hypertension [33]. 

The 2015 ESC/ERS guidelines indicated 13 parameters in the one-year mortality risk assessment tool for PAH, with three risk categories: low (<5% mortality rate), intermediate (5–10% mortality rate), and high (>10% mortality rate), respectively. The main objective with risk stratification is to achieve and maintain a low-risk profile [13]. Recently, the validity of this risk assessment algorithm and its abbreviated version, including only three parameters (WHO FC, 6MWD, and BNP/NT-proBNP), have been also tested in CTEPH patients in three studies from large European registries [14,15,16]. These studies also confirmed that achieving and maintaining a low-risk profile is associated with improved prognosis in CTEPH patients even if taking the effect of PEA into account. The results of present study demonstrated the utility of ESC/ERS risk assessment tool in the mortality prediction. However, PAPI model presents higher discriminative ability than ESC algorithm. Our findings suggest that hemodynamic risk evaluation of CTEPH patients remains crucial in prognostication. PAPI seems to be useful mortality predictor as compared with other recommended parameters in published risk prediction models and has an improved predictive ability compared with using mRAP alone or CI and PVR. 

Moreover, the present study demonstrated that the three parameters of 6MWD, RAA, and PAPI were closely linked to the mortality risk at baseline. Previously, Delacroix et al. based on PH-registry, COMPERA indicated BNP/NT-proBNP and WHO FC as independent determinants of survival in CTEPH [14]. Another study performed on the 237 patients enrolled in CHEST-2 study demonstrated that 6MWD and NT-proBNP concentration at baseline as well as change from baseline to follow-up were significantly independent predictors of survival [16]. Interestingly, an analysis of Swedish PAH Registry showed that WHO FC, 6MWD, NT-proBNP, RAP, CI, and SvO_2_ corresponded to the mortality risk at baseline [15]. When PAPI was used to stratify our CTEPH cohort into high-risk and low-risk groups, these groups varied significantly in 6MWD but were comparable in WHO FC. The association between WHO FC and NT-proBNP concentration with survival was not significant in present study.

This study is burdened by several limitations. The main limitation is its relatively small number of patients from a single center, which results from the low prevalence of CTEPH. This study concerns only inoperable CTEPH patients, and therefore, results cannot be applied to patients with operable CTEPH patients. Another limitation is that we could not validate the applicability of PAPI in risk prognostication using other larger cohorts. However, we strongly believe that these limitations have a relatively limited impact on the overall results and conclusions of this study.

## 5. Conclusions

In our cohort of patients with inoperable CTEPH, low PAPI (<3.9) was indicated as a significant independent predictor of mortality. Moreover, PAPI appears to be stronger risk indicator than mRAP alone or ESC 2015 risk assessment tool. However, future studies are needed to define the value of PAPI for risk stratification in CTEPH patients with inoperable disease for the decision-making process regarding the treatment options.

## Figures and Tables

**Figure 1 jcm-11-02735-f001:**
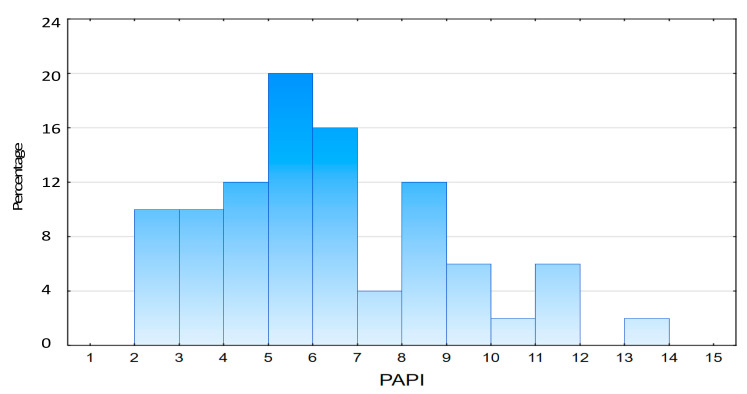
Distribution of PAPI among study population. Abbreviations: PAPI, pulmonary artery pulsatility index.

**Figure 2 jcm-11-02735-f002:**
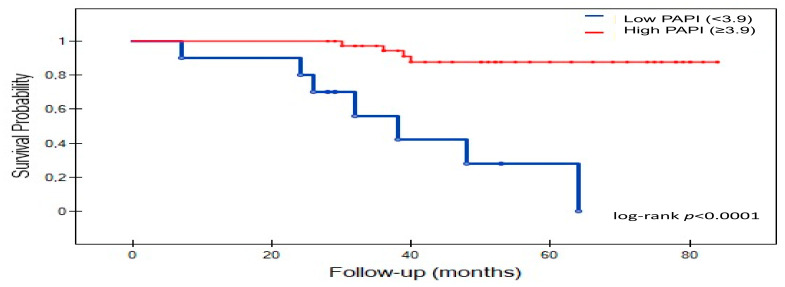
Kaplan−Meier survival curves analysis for low and high-PAPI group. Abbreviations: PAPI, pulmonary artery pulsatility index.

**Figure 3 jcm-11-02735-f003:**
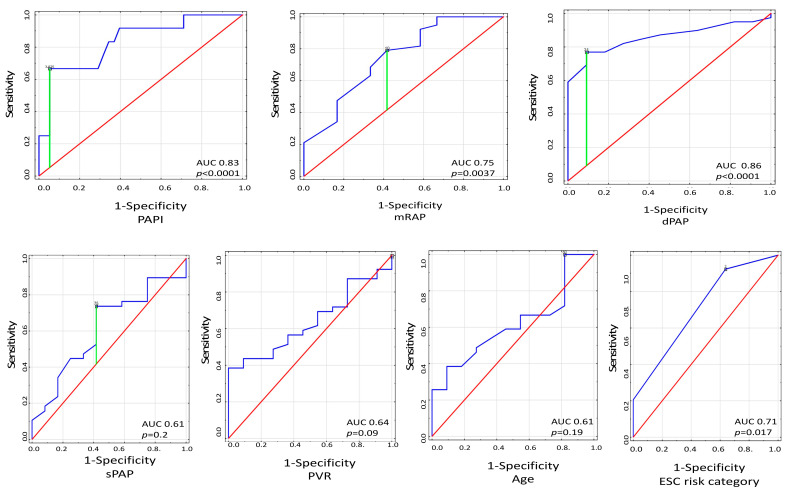
Receiver-operating characteristics curves for the ability of PAPI, mRAP, sPAP, dPAP, ESC risk model, and age to predict mortality risk. Red line represents reference line. Abbreviations: dPAP, diastolic pulmonary arterial pressure; ESC, European Society of Cardiology; mRAP, mean right atrial pressure; PAPI, pulmonary artery pulsatility index; PVR, pulmonary vascular resistance; sPAP, systolic pulmonary arterial pressure.

**Figure 4 jcm-11-02735-f004:**
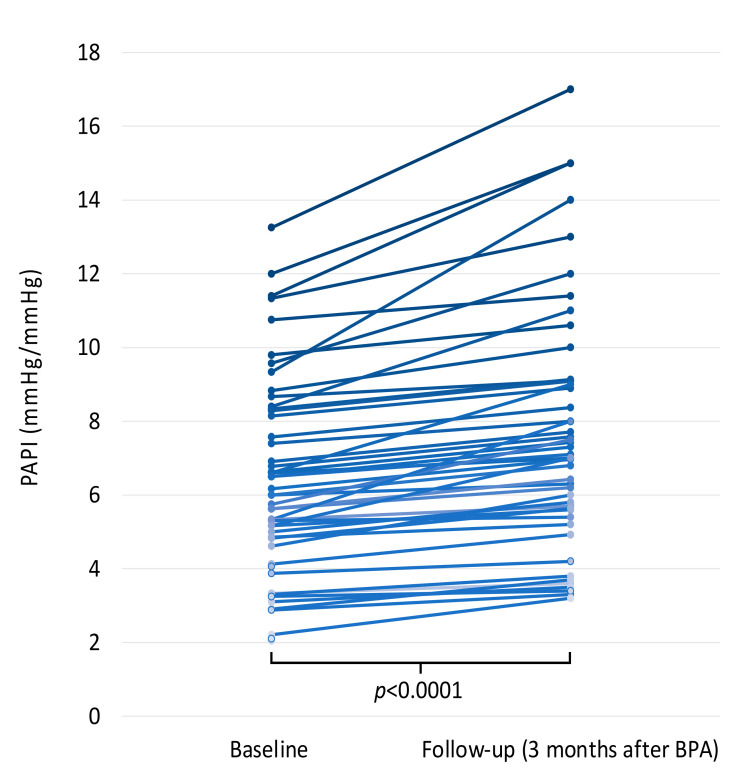
Change in PAPI values from baseline to follow-up 3 months after BPA treatment. Abbreviations: BPA, balloon pulmonary angioplasty; PAPI, pulmonary artery pulsatility index.

**Table 1 jcm-11-02735-t001:** Demographic and clinical characteristics of the study population according to pulmonary artery pulsatility index-based risk assessment.

Parameter	All *N* = 50, (%)	Low PAPI (<3.9) *N* = 10 (%)	High PAPI (≥3.9) *N* = 40 (%)	*p*-Value
Age (years), mean (SD)	64 (12.2)	63.4 (7.0)	64.3 (13.3)	0.78
Sex				0.72
female	30 (60)	6 (60)	24 (60)	
male	20 (40)	4 (40)	16 (40)	
BMI (kg/m^2^)	29.3 (7.0)	33.8 (5.3)	28.2 (7.1)	0.0066
WHO FC				0.0017
I	-	-	-	
II	9 (18)	-	9 (22.5)	
III	28 (56)	3 (30)	25 (62.5)	
IV	12 (24)	7 (70)	5 (12.5)	
Previous pulmonary embolism				0.14
yes	34 (68)	9 (90)	25 (62.5)	
no	16 (32)	1 (10)	15 (37.5)	
Coronary artery disease	10 (20)	3 (30)	7 (17.5)	0.41
Chronic obstructive pulmonary disease	7 (14)	2 (20)	5 (12.5)	0.62
Diabetes mellitus	11 (22)	4 (40)	7 (17.5)	0.2
Systemic arterial hypertension	35 (70)	8 (80)	27 (67.5)	0.7
Known thrombophilia	2 (4)	-	2 (5)	0.77
Chronic renal insufficiency	11 (22)	5 (50)	6 (15)	0.03
Anticoagulation therapy				
DOAC	33 (66)	6 (60)	27 (67.5)	0.72
VKA	17 (34)	4 (40)	13 (32.5)	0.72
6MWD (m), mean (SD)	308.2 (120)	236.2 (108.5)	328.2 (116.3)	0.03
NT-proBNP (pg/mL), mean (SD)	2296.4 (2939)	1566.8 (794.8)	2483.4 (3254.2)	0.51
ESC 2015 risk category				0.1
low (1)	8 (16)	-	8 (20)	
intermediate (2)	35 (70)	7 (70)	28 (70)	
high (3)	7 (14)	3 (30)	4 (10)	

Abbreviations: BMI, body mass index; BSA, body surface area; DOAC, direct oral anticoagulant; IQR, interquartile range; NT-proBNP, N-terminal brain natriuretic propeptide; SD, standard deviation; WHO-FC, World Health Organization Functional Class; VKA, vitamin K antagonist; 6MWD, 6-min walking distance; CTEPH, chronic thromboembolic pulmonary hypertension.

**Table 2 jcm-11-02735-t002:** Comparison of hemodynamic and echocardiographic data between low- and high-PAPI groups.

Parameter Mean (SD)	All *N* = 50	Low PAPI (<3.9) *N* = 10	High PAPI (≥3.9) *N* = 40	*p*-Value
Hemodynamic data				
sSAP (mmHg)	143.8 (26)	139.5 (26.8)	144.9 (26.1)	0.57
dSAP (mmHg)	85 (17.6)	77.6 (31.5)	87.5 (11.7)	0.22
mRAP (mmHg)	9.2 (4.6)	14.9 (4.4)	7.8 (2.8)	0.0001
sRVP (mmHg)	78 (17.3)	71.2 (18.5)	79.7 (16.8)	0.17
dRVP (mmHg)	6.0 (5.6)	9.2 (4.9)	5.2 (4.1)	0.018
edRVP (mmHg)	12.3 (5.1)	16.5 (4.6)	11.2 (4.7)	0.004
sPAP (mmHg)	79.7 (15.3)	79.1 (12.4)	79.9 (16.4)	0.089
dPAP (mmHg)	29.3 (7.4)	35.8 (5.3)	27.7 (7.0)	0.0012
mPAP (mmHg)	48.1 (9.2)	50.9 (8.5)	47.5 (9.4)	0.30
PAWP (mmHg)	10.1 (2.9)	11.2 (2.8)	9.9 (2.9)	0.066
PAPI (mmHg/mmHg)	6.4 (2.7)	3.0 (0.8)	7.2 (2.3)	< 0.0001
PVR (Wood units)	7.2 (3.2)	6.12 (2.5)	7.5 (3.3)	0.31
SVR (Wood units)	18 (7.0)	15 (6.2)	18.8 (7.1)	0.67
CO (L/min)	5.8 (1.6)	5.6 (1.5)	6.5 (1.7)	0.10
CI (L/min/m^2^)	3.1 (0.75)	3.0 (0.75)	3.3 (0.78)	0.36
SV (mL)	76.6 (24.5)	70.7 (33.1)	78 (22.1)	0.4
SvO_2_ (%)	67.6 (6.3)	66.5 (5.6)	67.9 (6.5)	0.55
SaO_2_ (%)	91.6 (3.6)	90.7 (4.12)	91.8 (3.4)	0.5
Echocardiographic data				
RAA (cm^2^)	27.4 (9.5)	34.3 (11.1)	25.7 (8.3)	0.026
RV free wall thickness (mm)	5.4 (0.9)	6 (0.7)	4.8 (1.1)	0.1
RV end-diastolic diameter (4 ch) (mm)	48.6 (9.6)	52.2 (10.1)	47.7 (9.4)	0.21
TAPSE (mm)	19 (4.9)	16.4 (4.0)	19.6 (8.5)	0.57
TRV max (m/s)	4.4 (0.6)	4.12 (0.43)	4.46 (0.61)	0.1
TVPG (mmHg)	82 (20.5)	71 (16.5)	85 (20.7)	0.21
Tricuspid regurgitation severity *n* (%)				0.9
Mild	5 (10)	-	5 (12.5)	
Moderate	40 (80)	8 (80)	32 (80)	
Severe	5 (10)	1 (10)	4 (10)	
Pulmonary trunk diameter (mm)	30.2 (5.7)	34.3 (2.1)	29.6 (5.8)	0.026
S’ wave (cm/s)	12.02 (3.4)	11.4 (4.3)	12.2 (3.2)	0.56

Abbreviations: CI, cardiac index; CO, cardiac output; dRVP, diastolic right ventricular pressure; dSAP, diastolic systemic arterial pressure; edRVP, end-diastolic ventricular pressure; mPAP, mean pulmonary arterial pressure; mRAP, mean right atrial pressure; PAWP, pulmonary artery wedge pressure; PVR, pulmonary vascular resistance; PAPI, pulmonary artery pulsatility index; RAA, right atrium area; RV, right ventricle; sRVP, systolic right ventricular pressure; sPAP, systolic pulmonary arterial pressure; SD, standard deviation; SaO_2_, arterial blood saturation; sSAP, systolic systemic arterial pressure; SV, stroke volume; SVR, systemic vascular resistance, SvO_2_, mixed venous saturation; TAPSE, tricuspid annular plane systolic excursion; TRV, tricuspid regurgitation velocity; TVPG, tricuspid valve pressure gradient.

**Table 3 jcm-11-02735-t003:** Univariate and multivariable Cox proportional hazards regression analysis.

	Univariate Cox Proportional Analysis			Multivariable Cox Proportional Analysis		
Variable	Hazard Ratio	95% CI	*p*-Value	Hazard Ratio	95% CI	*p*-Value
PAPI (per 1 unit)	0.48	0.32–0.72	0.0004	0.65	0.44–0.96	0.03
6MWD (per 1 m)	0.99	0.98–0.99	0.002	0.99	0.98–0.99	0.018
RAA (per 1 cm^2^)	1.1	1.04–1.17	0.001	1.07	1.00–1.15	0.03
mRAP (per 1 mmHg)	1.2	1.1–1.4	0.001	ns		
TAPSE (per 1 mm)	0.86	0.75–0.99	0.03	ns		
WHO FC (III/IV vs. I/II)	1.4	0.54–2.75	0.006	ns		

Abbreviations: mRAP, mean right atrial pressure; ns, non-significant; PAPI, pulmonary artery pulsatility index; RAA, right atrium area; TAPSE, tricuspid annular plane systolic excursion; WHO FC, World Health Organization Functional Class; 6MWD, six-minute walking distance.

## Data Availability

The data that support the findings of this study are available from the corresponding author upon reasonable request.

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
