# Peer review of "Prognostic Value of Pulmonary Artery Pulsatility Index in Right Ventricle Failure-Related Mortality in Inoperable Chronic Thromboembolic Pulmonary Hypertension"

_jcm, 2022, doi:10.3390/jcm11102735_

Round 1
Reviewer 1 Report
In this study, lower PAPI was associated to poor outcome of inoperable CTEPH. The authors concluded that PAPI might be useful predictor of survival in inoperable CTEPH. The point of view is interesting, however, a number of aspects of the manuscript need attention.
Major
- The most alarming point of this study is ambiguity of study design. The prognosis of inoperable CTEPH has evolved with the emergence of BPA and CTEPH approved drug. In this study, the authors analyzed baseline clinical characteristics. The authors never mentioned about treatments (BPA, medication, etc…) which would influence the prognosis. All patients received the appropriate BPA treatment or medical therapy? Since treatment for inoperable CTEPH has already been established, it is unreasonable to discuss only baseline hemodynamics.
- Small sample size.
- If patients were classified into Low PAPI group, and High PAPI group according to PAPI 3.9, the method of calculating the PAPI cut-off value as 3.9 should be described first in the Results section.
Author Response
We are deeply grateful to the Reviewer for taking the time to provide valuable comments.
A point-by-point response to the Reviewer’s comments is below.
Comment 1: The most alarming point of this study is ambiguity of study design. The prognosis of inoperable CTEPH has evolved with the emergence of BPA and CTEPH approved drug. In this study, the authors analyzed baseline clinical characteristics. The authors never mentioned about treatments (BPA, medication, etc…) which would influence the prognosis. All patients received the appropriate BPA treatment or medical therapy? Since treatment for inoperable CTEPH has already been established, it is unreasonable to discuss only baseline hemodynamics. Response: Thank you for this comment. In our cohort, each patient with inoperable CTEPH received medical therapy with soluble guanylate cyclase stimulator, riociguat – the only drug approved for treating CTEPH. Riociguat was initiated within 3 months of diagnosis. Moreover, patients were offered balloon pulmonary angioplasty (BPA) procedures with detailed information about the potential risks and benefits of this intervention, and each patients underwent series of BPA in our center. Notably, in this study we identified a baseline low-risk profile in inoperable CTEPH defined by PAPI ≥3.9 and showed that patients with this low-risk profile had better survival than patients with high-risk profile (PAPI<3.9). However, this is the preliminary study, our future goal is to determine if changes in the PAPI-derived risk category, regardless of the direction, may be linked with the prognosis. Authors are now working on such study protocol.
Comment 2: Small sample size. Response: Thank you. We have included only inoperable patients, excluding patients with the proximal CTEPH or persistent pulmonary hypertension after pulmonary endarterectomy. The estimated CTEPH frequency is about 0.9 and 3.2 cases per million, while inoperable CTEPH prevalence is 30-45% of all CTEPH cases [1]. We have performed a power analysis during study design and the estimated study group was calculated 50 cases with a test’s power at the level 0.8.
Reference 1. Yandrapalli S., Tariq S., Kumar J.,et al. Chronic Thromboembolic Pulmonary Hypertension: Epidemiology, Diagnosis, and Management. Cardiol. Rev. 2018;26:62–72.
Comment 3: If patients were classified into Low PAPI group, and High PAPI group according to PAPI 3.9, the method of calculating the PAPI cut-off value as 3.9 should be described first in the Results section. Response: Thank you for pointing this out. We have added a clear description in the statistical methods section, as we consider more appropriate than results section. “PAPI calculated from hemodynamic data was categorized into ‘low’ and ‘high’ groups with the chi-square automated interactions detector algorithm as described previously [19, 23]. This is a multi-way splitting decision tree used for identifying the predictors or explaining an outcome based on the adjusted Bonferroni testing or determining the optimal cut-off values for the predictors. The generalized structural equation model was used for confirmatory analyses, with Binomial and Weibull chosen as the underlying distributions for assessing the occurrence of death (1: yes; 0: no) and time to death (months), respectively. Beyond the PAPI, the predictors also involved demographics, clinical data, and other hemodynamic parameters. The results were presented based on a backward elimination procedure in model-selection, which was in turned based on the unadjusted analyses. The accuracy of the derived PAPI cut-off was subsequently evaluated with the receiver operating characteristics (ROC) curve.”
Reviewer 2 Report
This is a clinical study, which aimed to evaluate whether PAPI might be useful for mortality prediction in inoperable CTEPH. The authors concluded that low PAPI (<3.9) was indicated as significant independent predictor of mortality in inoperable CTEPH patients, and that PAPI appears to be a stronger risk indicator than mRAP alone or ESC 2015 risk assessment tool. This is an important issue, and this reviewer considers that the authors well performed the present study. This reviewer has some comments as described below.
Major comments:
- The authors measured cardiac output, but did not describe how. Did the authors perform by Fick or a thermodilution method?
- Table 1, previous symptomatic pulmonary embolism. There are 2 lines of data. It seems that one is “yes”, and the other is “no”. They should be corrected. Further, “)” was lacked after “25(62.5”.
- Figure 1. What was red curve? If it is not statistically scientific, it should be deleted.
- Figure 2 and other results. Where do patients with PAPI=3.9 go?
Minor comment:
- There were several typos. For example, in page 2, line 5, hyphen was in red. In page 10, in conclusion, “a” was lacked in front of “stronger risk indicator”.
Author Response
Comment 1: The authors measured cardiac output but did not describe how. Did the authors perform by Fick or a thermodilution method? Response: Thank you for pointing this out. Cardiac output was evaluated using the thermodilution method- we have this information in the methods section (line 97).
Comment 2: Table 1, previous symptomatic pulmonary embolism. There are 2 lines of data. It seems that one is “yes”, and the other is “no”. They should be corrected. Further, “)” was lacked after “25(62.5”.
Response: Thank you for this comment. We have corrected these data according to your advice.
Comment 3: Figure 1. What was red curve? If it is not statistically scientific, it should be deleted.
Response: Thank you. We have corrected it according to your suggestion – red line was removed
Figure 1. The distribution of PAPI.
Comment 4: Figure 2 and other results. Where do patients with PAPI=3.9 go?
Response: Thank you for highlighting this. The cut-off point was 3.9, and patients with PAPI equal or higher 3.9 were classified as high PAPI group. We have corrected manuscript (line 164) as well as each table and figure 2, accordingly.
Figure 2. The Kaplan-Meier survival curves analysis for PAPI groups.
Comment 5: There were several typos. For example, in page 2, line 5, hyphen was in red. In page 10, in conclusion, “a” was lacked in front of “stronger risk indicator”.
Response: Thank you. We have corrected all typos and technical errors pointed out by the Reviewer.

Reviewer 3 Report
The manuscript presented by Sławek-Szmyt et al. explored the prognostic role of pulmonary artery pulsatility index in patients affected by inoperable chronic thromboembolic pulmonary hypertension. The results of this monocentric study showed that a low value of PAPI was a strong negative prognostic marker in this cohort of patients, identifying 3.9 as a potential cut-off in this disease. The idea of the Authors of exploring this parameter in stratifying the prognostic risk of patients affected by chronic thromboembolic pulmonary hypertension was interesting and could add some value to the literature in this field. However, I personally find many limitations in this present work.
Major points to be addressed:
-It is definitely not clear how the multivariable model was performed. Authors say in the statistical method that an univariable model and a multivariable was performed. However, it is presented only table 3 where the authors show that PAPI, RAA and 6MWD were significantly associated with mortality. Were them the only parameters significantly to the prognosis at the univariable analysis? Authors should report the complete model. Furthermore, in the text it is reported “multivariable logistic regression model”, while in the table it is reported Cox regression model, which would be correct for the present analysis. Please clarify it.
-In the section methods it is said that all patients were judged inoperable and patients were assigned to medical treatment and staged BPA procedures. It is however not reported in the manuscript how many patients were taking riociguat, or how many patients had BPA procedure. In case there are differences of treatment in the cohort with low PAPI vs high PAPI it has to be considered as a potential confounding and must be included in the model.
-Twenty potentially eligible patients were excluded from the present analysis for different reasons. Considering the low sample size of the study cohort it is a consistent number. Authors must provide a supplementary table with the characteristics of the excluded patients to exclude potential selection bias.
-Were patients furtherly assessed by RHC during follow-up? It would be interesting to see if a variation of PAPI due to medical treatment or BPA could be also predictive of events.
-Thermodilution was adopted to evaluate CO. It is not mentioned how many patients had significant tricuspid regurgitation, which would limit the accuracy of the technique.
Minor points:
-English language must be adequately revised. Furthermore, there are many typing mistakes that should be rephrased. Some examples: introduction, page 1 lines 8-11 “Available treatment options for these CTEPH patients include targeted medical therapy and interventional technique – percu[1]taneous balloon pulmonary angioplasty (BPA), which has become a promising treatment medical treatment in CTEPH is limited”; pag.2 line 5, please remove -before “parameter”. This is not an extensive list, there are some others. Please correct them.
-I did not understand the meaning of the percentages in the headings of table 2.
-I assume that the symbol * the significant parameters listed in the table 2. However, many parameters which result significant are not marked.
Author Response
Rewiever 3
Comment 1: It is definitely not clear how the multivariable model was performed. Authors say in the statistical method that an univariable model and a multivariable was performed. However, it is presented only table 3 where the authors show that PAPI, RAA and 6MWD were significantly associated with mortality. Were them the only parameters significantly to the prognosis at the univariable analysis? Authors should report the complete model. Furthermore, in the text it is reported “multivariable logistic regression model”, while in the table it is reported Cox regression model, which would be correct for the present analysis. Please clarify it.
Response: We are grateful for this comment. We have clarified the implemented methods in statistical analysis section and added the results of univariate Cox proportional analysis on Table 3.
“Survival was analyzed both at baseline and follow-up using the Kaplan–Meier method with log-rank test. Predictors for survival were determined with univariable and multivariable Cox proportional regression analyses”.
Table 3. Univariate and multivariable Cox proportional hazards regression analysis.
|
|
Univariate Cox proportional analysis |
Multivariable Cox proportional analysis |
|
||||
|
Variable |
Hazard ratio |
95% CI |
p value |
Hazard ratio |
95% CI |
p value |
|
|
PAPI (per 1 unit) |
0.48 |
0.32-0.72 |
0.0004 |
0.65 |
0.44-0.96 |
0.03 |
|
|
6MWD (per 1 m) |
0.99 |
0.98-0.99 |
0.002 |
0.99 |
0.98-0.99 |
0.018 |
|
|
RAA (per 1 cm2) |
1.1 |
1.04-1.17 |
0.001 |
1.07 |
1.00-1.15 |
0.03 |
|
|
mRAP (per 1 mmHg) |
1.2 |
1.1-1.4 |
0.001 |
ns |
|
|
|
|
TAPSE (per 1 mm) |
0.86 |
0.75-0.99 |
0.03 |
ns |
|
|
|
|
WHO FC (III/IV vs I/II) |
1.4 |
0.54-2.75 |
0.006 |
ns |
|
|
|
Abbreviations: mRAP– mean right atrial pressure; PAPI- pulmonary artery pulsatility index; RAA – right atrium area; TAPSE– tricuspid annular plane systolic excursion; WHO FC– World Health Organization Functional Class; 6MWD- six-minute walking distance
Comment 2: Twenty potentially eligible patients were excluded from the present analysis for different reasons. Considering the low sample size of the study cohort it is a consistent number. Authors must provide a supplementary table with the characteristics of the excluded patients to exclude potential selection bias.
Response: Thank you for pointing this out. We have taken this comment to heart and created a new supplementary table 1 presenting the characteristics of the excluded patients (attached also below). Moreover, we have performed a power analysis during study design and the estimated study group was calculated 50 cases with a test’s power at the level 0.8.
Supplementary table 1. Characteristics of inoperable CTEPH patients excluded from the study.
|
Patient ID |
Sex |
Age at diagnosis (years) |
Cause of exclusion |
Cause of death |
|
1 |
F |
70 |
Lack of consent |
|
|
2 |
M |
83 |
Non-CTEPH related death |
Colon cancer |
|
3 |
F |
67 |
Non-CTEPH related death |
Abdominal surgery-related complications |
|
4 |
F |
89 |
Non-CTEPH related death |
Sepsis |
|
5 |
F |
27 |
CTEPH persistent after PEA |
- |
|
6 |
F |
57 |
CTEPH persistent after PEA |
|
|
7 |
M |
73 |
Non-CTEPH related death |
Orthopedic surgery related complication |
|
8 |
F |
36 |
Lack of follow-up |
|
|
9 |
M |
66 |
Lack of consent |
|
|
10 |
F |
78 |
Lack of follow-up |
|
|
11 |
M |
59 |
CTEPH persistent after PEA |
|
|
12 |
M |
67 |
CTEPH persistent after PEA |
|
|
13 |
F |
70 |
Lack of consent |
|
|
14 |
M |
53 |
Lack of consent |
|
|
15 |
M |
55 |
CTEPH persistent after PEA |
|
|
16 |
M |
89 |
Non-CTEPH related death |
COVID-19 |
|
17 |
M |
70 |
Non-CTEPH related death |
Sepsis |
|
18 |
F |
74 |
Non-CTEPH related death |
Sepsis |
|
19 |
M |
59 |
Non-CTEPH related death |
Colon cancer |
|
20 |
F |
61 |
Unknown cause of death |
|
Comment 3: In the section methods it is said that all patients were judged inoperable and patients were assigned to medical treatment and staged BPA procedures. It is however not reported in the manuscript how many patients were taking riociguat, or how many patients had BPA procedure. In case there are differences of treatment in the cohort with low PAPI vs high PAPI it has to be considered as a potential confounding and must be included in the model.
Response: Thank you for this comment. In our cohort, each patient with inoperable CTEPH received medical therapy with soluble guanylate cyclase stimulator, riociguat – the only drug approved for treating CTEPH. Riociguat was initiated within 3 months of diagnosis. Moreover, patients were offered balloon pulmonary angioplasty (BPA) procedures with detailed information about the potential risks and benefits of this intervention, and each patients underwent series of BPA in our center.
Comment 4: Were patients furtherly assessed by RHC during follow-up? It would be interesting to see if a variation of PAPI due to medical treatment or BPA could be also predictive of events.
Response: Thank you for this comment. We have added a new figure (figure 4- attached also below) presenting PAPI values assessed 3-months after interventional treatment of CTEPH with BPA.
Figure 4. Change in PAPI values from baseline to follow-up 3 month after BPA treatment.
Comment 5: Thermodilution was adopted to evaluate CO. It is not mentioned how many patients had significant tricuspid regurgitation, which would limit the accuracy of the technique.
Response: Thank you for this comment. Although there has been debate regarding the validity of thermodilution in patients with low CO or severe tricuspid regurgitation (1-3) other studies14,30 have reported that agreement between thermodilution and direct Fick (using measured V̇O2) estimates remains robust even in these scenarios (4-6). However, we have added detailed data regarding tricuspid regurgitation severity in Table 2 (attached also below). We strongly believe that in our cohort the impact of severe tricuspid regurgitation on CO results were not statistically significant. It should be emphasized that, the majority of patients had CO values higher than 4.5 l/min.
References:
- Guyton AC, Jones CE, Coleman TG. Cardiac Output and Its Regulation. 2nd ed. Philadelphia, PA: Saunders; 1973.
- Cigarroa RG, Lange RA, Williams RH, Bedotto JB, Hillis LD. Underestimation of cardiac output by thermodilution in patients with tricuspid regurgitation. Am J Med. 1989;86(4):417-420.
- van Grondelle A, Ditchey RV, Groves BM, Wagner WW Jr, Reeves JT. Thermodilution method overestimates low cardiac output in humans. Am J Physiol. 1983;245(4):H690-H692.
- Hoeper MM, Maier R, Tongers J, et al. Determination of cardiac output by the Fick method, thermodilution, and acetylene rebreathing in pulmonary hypertension. Am J Respir Crit Care Med. 1999;160(2):535-541.
- Yung GL, Fedullo PF, Kinninger K, Johnson W, Channick RN. Comparison of impedance cardiography to direct Fick and thermodilution cardiac output determination in pulmonary arterial hypertension. Congest Heart Fail. 2004;10(2)(suppl 2):7-10.
- Opotowsky AR, Hess E, Maron BA, et al. Thermodilution vs Estimated Fick Cardiac Output Measurement in Clinical Practice: An Analysis of Mortality From the Veterans Affairs Clinical Assessment, Reporting, and Tracking (VA CART) Program and Vanderbilt University. JAMA Cardiol. 2017;2(10):1090-1099.
Table 2.
|
Tricuspid regurgitation severity n (%) Mild Moderate Severe
|
All
5 (10) 40 (80) 5 (10) |
Low PAPI (<3.9)
- 8 (80) 1 (10) |
High PAPI (≥3.9)
5 (12.5) 32 (80) 4 (10) |
p value 0.9 |
Comment 6: English language must be adequately revised. Furthermore, there are many typing mistakes that should be rephrased. Some examples: introduction, page 1 lines 8-11 “Available treatment options for these CTEPH patients include targeted medical therapy and interventional technique – percu[1]taneous balloon pulmonary angioplasty (BPA), which has become a promising treatment medical treatment in CTEPH is limited”; pag.2 line 5, please remove -before “parameter”. This is not an extensive list, there are some others. Please correct them.
Response: We are grateful for this comment. We have corrected English grammatical errors.
Comment 7: I did not understand the meaning of the percentages in the headings of table 2.
Response: Removed.
Comment 8: I assume that the symbol * the significant parameters listed in the table 2. However, many parameters which result significant are not marked.
Response: Thank you. We have removed this marker from table 2.

Round 2
Reviewer 1 Report
Waiting for editor's reply.